# Nosocomial Coronavirus Disease 2019 during 2020–2021: Role of Architecture and Ventilation

**DOI:** 10.3390/healthcare12010046

**Published:** 2023-12-25

**Authors:** Martin Martinot, Mahsa Mohseni-Zadeh, Simon Gravier, Ciprian Ion, Magali Eyriey, Severine Beigue, Christophe Coutan, Jean-Claude Ongagna, Anais Henric, Anne Schieber, Loic Jochault, Christian Kempf

**Affiliations:** 1Infectious Diseases Department, Hôpitaux Civils de Colmar, 68000 Colmar, France; mahsa.mohseni@ch-colmar.fr (M.M.-Z.); simon.gravier@ch-colmar.fr (S.G.); ciprian.ion@gmail.com (C.I.); severine.beigue@ch-colmar.fr (S.B.); 2Clinical Research Department, Hôpitaux Civils de Colmar, 68000 Colmar, France; magali.eyriey@ch-colmar.fr (M.E.); jean-claude.ongagna@ch-colmar.fr (J.-C.O.); anais.henric@ch-colmar.fr (A.H.); anne.schieber@ch-colmar.fr (A.S.); christiankempf@yahoo.fr (C.K.); 3Technical Department, Hôpitaux Civils de Colmar, 68000 Colmar, France; christophe.coutan@ch-colmar.fr; 4Medical Information Service, Hôpitaux Civils de Colmar, 68000 Colmar, France; loic.jochault@ch-colmar.fr

**Keywords:** nosocomial COVID-19, SARS-CoV-2, ventilation, healthcare settings, architecture, single room

## Abstract

Nosocomial coronavirus disease 2019 (COVID-19) is a major airborne health threat for inpatients. Architecture and ventilation are key elements to prevent nosocomial COVID-19 (NC), but real-life data are challenging to collect. We aimed to retrospectively assess the impact of the type of ventilation and the ratio of single/double rooms on the risk of NC (acquisition of COVID-19 at least 48 h after admission). This study was conducted in a tertiary hospital composed of two main structures (one historical and one modern), which were the sites of acquisition of NC: historical (H) (natural ventilation, 53% single rooms) or modern (M) hospital (double-flow mechanical ventilation, 91% single rooms). During the study period (1 October 2020 to 31 May 2021), 1020 patients presented with COVID-19, with 150 (14.7%) of them being NC (median delay of acquisition, 12 days). As compared with non-nosocomial cases, the patients with NC were older (79 years vs. 72 years; *p* < 0.001) and exhibited higher mortality risk (32.7% vs. 14.1%; *p* < 0.001). Among the 150 NC cases, 99.3% were diagnosed in H, mainly in four medical departments. A total of 73 cases were diagnosed in single rooms versus 77 in double rooms, including 26 secondary cases. Measured air changes per hour were lower in H than in M. We hypothesized that in H, SARS-CoV-2 transmission was favored by short-range transmission within a high ratio of double rooms, but also during clusters, via far-afield transmission through virus-laden aerosols favored by low air changes per hour. A better knowledge of the mechanism of airborne risk in healthcare establishments should lead to the implementation of corrective measures when necessary. People’s health is improved using not only personal but also collective protective equipment, i.e., ventilation and architecture, thereby reinforcing the need to change institutional and professional practices.

## 1. Introduction

Nosocomial coronavirus disease 2019 (COVID-19) is a major health threat for inpatients [1,2,3]. Although vaccination provides protective effects, the surge in Omicron infections was associated with a significant increase in hospital-onset severe acute respiratory syndrome coronavirus 2 (SARS-CoV-2) infections [4]. Notably, nosocomial infections have higher mortality rates than community-acquired diseases, particularly in the elderly or patients who have suppressed immune systems [1,5,6]. Hence, this issue must be addressed to reduce mortality due to COVID-19. There is a continuum between droplets (particles > 5 µm) and particles (<5 µm) that are immediately respirable by exposed individuals, causing far-afield contamination [7], particularly in indoor settings [8]. Therefore, there is a risk for near-source as well as far-afield transmission caused by these particles, particularly within enclosed spaces and areas with inadequate ventilation [9,10,11]. Single rooms limit close contamination. Ventilation, and filtration procedures can reduce or remove the number of virus-laden aerosols and provide a determined volume of air changes per hour (ACH), reducing far-afield contamination among patients, healthcare workers (HCWs), and visitors [12,13]. Ventilation systems can be classified into natural and mechanical systems. Natural ventilation (NV) from outdoor air results in a low ACH, especially without regular aeration via opening of the windows, and it is present in households and older medical structures or long-term care facilities. New strategies in ventilation and building conception have been described following the COVID-19 pandemic to diminish viral transmission in healthcare settings; these modifications include increasing the ventilation rates, avoiding air recirculation, minimizing the number of people indoors [14], and using filtration and other purification techniques installed in the HVAC systems, mobile (high-efficiency particulate air) HEPA filtration units [8,15], or UV-based technologies [16].

Hôpitaux Civils de Colmar (Figure 1) is a tertiary hospital with two main establishments within the same geographical zone: a historical establishment (H) with multiple buildings built during the 20th century and having NV, except for toilets with exhaust fans (558 short-stay beds, 53% single rooms), and a modern establishment (M) opened in 2018 and having double-flow mechanical ventilation (MV), allowing fresh air to be injected into patients’ rooms (141 short-stay beds, 91% single rooms). The structural characteristics thus differ, mainly by a higher number of single rooms and enhanced ventilation in M. H is mainly dedicated to adults in medical and surgical departments, and M to patients in the pediatric and obstetrics and gynecology departments but also in the intensive care unit (ICU). Since the end of 2020 and the implementation of universal screening for hospitalized patients due to the COVID-19 pandemic, we aimed to retrospectively assess the burden of nosocomial COVID-19 (NC) and the impact of the type of building (ventilation) and the ratio of single/double rooms on the risk of NC.

## 2. Study Design

### 2.1. Methods and Patients

We retrospectively analyzed the data of all consecutive inpatients with COVID-19 in Hôpitaux Civils de Colmar from 1 October 2020 to 31 May 2021. COVID-19 was confirmed based on positive polymerase chain reaction (PCR) for SARS-CoV-2. Nosocomial COVID-19 was defined by a negative PCR result upon admission and a positive PCR finding >48 h after admission. This definition differs from the widely accepted definition of presumptive nosocomial COVID-19 (3–14 days) and a definite delay of >14 days.

### 2.2. Data Collection and Endpoints

The data collected from the computer-based patient records included sex, age, and mortality and discharge status. The time and place of acquisition of NC (H vs. M) and type of room (double against single) were collected. The ACH values were based on the “technical” data, and in a sample of rooms of H and M, ACH values were determined using an anemometer, including a hot wire anemometer (Testo^®^ 405i) and a flow rate cone (Testovent^®^ 410).

### 2.3. Statistical Analysis

Data were analyzed using SAS 9.4 software (SAS Institute, Cary, NC, USA).

Continuous variables were summarized as median and first and third quartiles (Q1 and Q3) and compared with the help of the Wilcoxon rank sum test. Categorical data were compared using the Chi-square. Mortality rate curves with 95% confidence intervals and hazard ratios were determined using the nonparametric Kaplan–Meier method. Survival curves were determined using the Kaplan–Meier method, censoring patients at day 50 post-admission or at the date of last news, whichever occurred first. Furthermore, patients who were transferred to another hospital or to a long-term care facility were followed up with through phone calls at least 50 days post-admission. For patients who were discharged to return home, no follow-up was conducted.

## 3. Results

During the study period, 33,718 patients were hospitalized, with an average hospital stay of 5.4 days (d), including 25,038 patients in H (74.3%) and 8680 in M (25.7%) with an average length stay of 5.5 and 4.7 d, respectively (Table 1). Overall, 1020 patients presented with COVID-19, which included 150 (14.7%) with NC. Nosocomial infection occurred at a median delay of 12 days (Q1: 7 d, Q3: 19 d). When comparing the 150 patients with NC to 870 with non-nosocomial COVID-19, patients with NC were older (79 vs. 72 years; *p* < 0.001) and had a higher mortality risk (32.7% vs. 14.1%; *p* < 0.001) than non-nosocomial cases (Figure 2). A total of 149 (99.3%) cases were acquired in H and 1 (0.7%) in M, showing a significant difference when compared with the number of admissions during the study period of H (25,038 patients) and M (8680 patients) (*p* < 0.0001). Most NC cases (98/150) were acquired in four medical departments within two buildings of H characterized by a high ratio of double rooms. A total of 73 (48.7%) patients were diagnosed in single rooms, and 77 (51.3%) were diagnosed in double rooms, including 26 secondary cases diagnosed. Table 2 shows patients’ characteristics and the site of infections.

ACH was determined in 10 rooms in H situated in three different buildings and in four rooms in M. In H, the mean ACH in building 1 was 1.1 volume/h in double rooms (r 0.89–1.27; room volume 55.8 m^3^; n = 3) and 2.01 volume/h in single rooms (r 1.9–2.15; room volume 25.7 m^3^; n = 3), 0.69 volume/h in building 2 (r 0.54–0.84; room volume 40.2 m^3^; n = 3 and 69.7 m^3^; n = 1), and 0.8 volume/h in building 3 (r 0.77–0.82; room volume 51.4 m^3^ and 67.6 m^3^). In M, the mean ACH was 1.65 vol/h (r 1.38–1.86; volume 61.41 m^3^; n = 2 and volume 41 m^3^; n = 2).

## 4. Discussion

In this study, NC cases accounted for 14.7% of inpatients diagnosed with COVID-19. Patients with NC who were older had a higher mortality rate of 32.7%, which was much higher than that in community-acquired cases, as previously described [1,5]. The rate of NC differed by period of stay and hospital [1], which shows the need to consider different factors when assessing NC. The study period, circulating variants, rates of immunization, and type of healthcare settings are key elements. In particular, modern healthcare settings with single rooms and MV cannot be compared with old healthcare settings. In this study, we assessed the role of architecture (single versus double room and ventilation) by assessing the site of infection (H vs. M healthcare facility), and the results indicated that M had a significantly lower rate (1%) of NC than H (99%). A single NC was diagnosed in M for a patient transferred to the room from a surgical department of H 48 h before performing the test; thus, even in this case, an acquisition in H seems probable. These results underscored the potential benefits of modern medical structures with single rooms and MV and are in accordance with new data on airborne pathogen transmission [11]. Half of the patients with NC were hospitalized in double rooms, with secondary cases diagnosed in 26 patients, probably via short-range contamination, as previously described [17,18]. The rate was, however, still high in patients who were infected in single or double rooms in the absence of identification of infected neighbors. For at least some of these cases without an unidentified infected source, a long-range contamination via virus-laden aerosols through corridors might be suspected, as reported by similar studies [19]. Although acquisition via HCWs or visitors cannot be ruled out, wearing a mask was mandatory for HCWs and visitors, and PPE was similarly recommended in M and H.

Low ACH due to NV in a department with a high rate of patients infected with COVID-19 (clusters), especially without regular ventilation in winter with low outdoor temperatures, may have contributed to these cases. Interestingly, in Park et al.’s study, aerosol contamination was favored by the fact that in winter, the windows were closed and doors were opened, allowing for contamination through the corridor [19]. The source of nosocomial infections is frequently unknown among airborne viral agents, and a high incidence of asymptomatic, pauci-symptomatic, or pre-symptomatic infections [20] makes the implementation of transmission-based precautions nearly impossible [8,21], emphasizing the potential interest of universal precautions integrating the airborne risk [12]. In addition to Park’s clinical study [19], the influence of MV and NV has already been emphasized in studies evaluating RNA detection, which is more common in healthcare settings with NV than in those with MV [13,22].

We did not determine the precise ACH value for all rooms and based our general ACH value on “technical” data. Thereafter, ACH in H was estimated to be ≤1 volume/h and approximately 2 volume/h in M. However, we determined precise ACH values in 14 rooms: 10 in H in three different buildings and 4 in M. In H, one the most affected buildings had 76 places (16 single rooms and 30 double rooms). The mean ACH was 1.1 volume/h in double rooms but higher in single rooms at 2.01 volume/h. We also checked the ACH in two other buildings of H: the oldest building, with a mean ACH of 0.69 volume/h, and the newest building of H, with a mean ACH of 0.8 volume/h. In M, where the patients’ rooms were more uniform, the mean ACH was 1.65 volume/h. These results were thus quite similar to the technical value, with a mean value ranging from 0.69 to 1.1 volume/h in H and 1.65 in M near the theorical value of 1 and 2, respectively. In M, the ACH was lower than the 6–12 recommended to prevent airborne infections in new healthcare structures. However, rooms in M had two other advantages: a double-flow MV, allowing for fresh air from the outside to enter rather than from the corridor, and a higher number of single rooms. This point could suggest that lower ACH values, associated with other architectural improvements such as exclusive single rooms, could be efficient in preventing airborne infections, with benefits in terms of energy and cost.

Ventilation and architecture (including single rooms) appear as key elements to prevent nosocomial airborne infections, and this study highlights the fact that nosocomial COVID-19 is easier to transmit in old settings without MV. Although it is difficult to assess the efficiency of each corrective measure, HCWs should be aware of these risks to implement corrective measures, especially in old healthcare settings. These corrective methods could include the integration of CO_2_ captors monitoring CO_2_ in medical departments, allowing alerts for levels >700 ppm, for example [23], and alerting HCWs to increase ACH by opening the windows. Modelization with the help of aerosol scientists of the natural airflow within a department is important to direct airflows “from clean to less clean” including when opening door and windows [12,19]. Moreover, adjusting exhaust fans to improve ACH can also be an easy way to ensure the best ACH. In the case of departments with low ACH mobile filtration units, the use of UV-based technologies could be discussed. The COVID-19 pandemic was also changed due to the high level of immunization among the population and the presence of the Omicron variant. Therefore, the most efficient collective protective equipment (MV with high ACH, HEPA mobile units, or UV-based technologies) should focus on departments accepting highly susceptible patients, such as those who are highly immunosuppressed and those who are critically ill, as well as departments treating patients with transmissible infections, such as infectious disease units, or departments or collective zones with a high number of inpatients, such as emergency wards or collective rooms in nursing homes. These modifications are necessary in daily practice, and they are critical during pandemics, when viral transmission is at a high risk, and there is an increased concentration of potentially infected patients in healthcare settings.

This study has several limitations. We defined nosocomial COVID-19 as being diagnosed >48 h after hospitalization rather than the widely accepted definition of presumptive nosocomial COVID-19 (3–14 days) and a definite delay of >14 days. However, all inpatients with NC had a negative PCR upon admission. Moreover, the median delay of acquisition was quite long (12 days). The difference in activities performed in H and M, with a longer hospital stay in H, and the fact that older patients are more susceptible to symptomatic COVID-19 is a clear limitation of this work. The four medical departments in H that were most affected were those which accepted a high number of patients from the emergency departments, usually those with a longer hospital stay. Although isolated from non-COVID departments, medical COVID units were located in H, except for the COVID ICU (located in M). These elements may have favored a higher density of virus in H than in M. A high concentration of infected patients within poorly ventilated spaces favors far-afield transmission [24]. Finally, the occurrence of clusters led to screening campaigns in departments, with NC favoring the diagnosis of nosocomial asymptomatic cases and a better awareness of nosocomial risk in physicians in H. Nevertheless, such a difference in patients with NC, with nearly no cases in M, underlines the potential importance of MV and single rooms in a context of missing real-live data. Other architectural characteristics are important, such as the position of beds in double rooms and the circulation of airflow in the rooms and departments, but these are complex elements to analyze, especially in H, which has different buildings built between 1937 and the beginning of the twenty-first century; thus, the surfaces of rooms and departments are very different. This study focused on preventing the transmission of airborne viruses. However, immunization is also important to prevent clusters, and inpatients’ vaccination status should be screened upon admission to implement supplementary vaccine doses when required [25].

## 5. Conclusions

Corrective measures to prevent airborne nosocomial infections have an impact that is difficult to assess. These are simple, even though they require a radical change in HCWs’ knowledge of “safe ventilation”. This study emphasized on the burden of NC during the winter of 2020–2021. The need for new guidelines on ventilation and single room ratios to prevent airborne infection in healthcare institutions is highlighted. In the future, infection prevention and control staff and aerosol scientists should always be included while designing healthcare facilities, and healthcare workers should integrate safe ventilation into standard precautions to prevent airborne infections. The use of single rooms and ventilation optimization, which can decrease the risk of far-afield infection (such as COVID-19) caused by airborne pathogens, should always be discussed in healthcare settings in epidemic waves. People’s health is improved using collective protective equipment, i.e., ventilation and architecture, thereby reinforcing the need to change institutional and professional practices.

## Figures and Tables

**Figure 1 healthcare-12-00046-f001:**
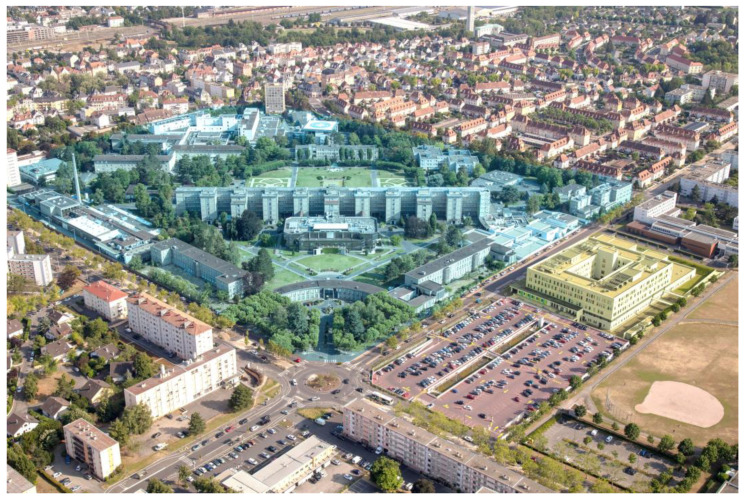
An aerial view of Hôpitaux Civils de Colmar with its two main geographical zones: a historic hospital (H in blue) at the center with buildings commonly built during the 20th century and having natural ventilation except in toilets with exhaust fans (558 beds, 53% are single rooms, and ACH of <1); and a modern hospital (M in yellow) with a square building to the right of the image that was opened in 2018 (141 short-stay beds, 91% are single rooms, with MV and ACH of >2).

**Figure 2 healthcare-12-00046-f002:**
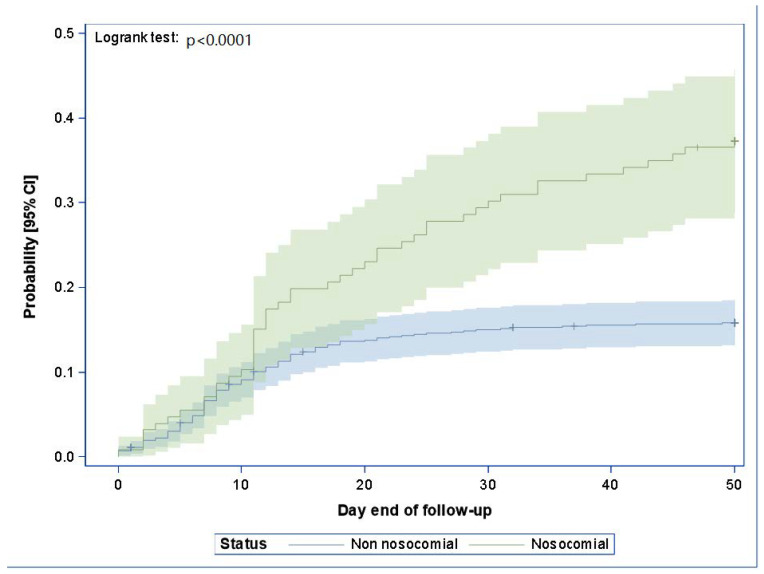
Death rate curve–product limit estimate (Kaplan–Meier) with two-sided confidence interval and number of patients at risk. Day 0 is the day of hospitalization. Patients alive at the end of the follow-up were excluded at the date of last information obtained.

**Table 1 healthcare-12-00046-t001:** Medical and surgical departments of H and M with the number of hospitalizations and average hospital stay (days). The four departments with the most NC cases are displayed in green.

Hospital H	Number of Hospitalizations	Average Length of Stay (Days)
Senology and plastic general, vascular, urology, ophthalmology, odontology, ENL surgery, and hepatogastroenterology departments	6524	3.1
ICU (neurosurgery/orthopedic and traumatological) and pain center	302	5.4
Endocrinology cardiology Nephrology	3038	6.2
Gerontology	449	12.9
Radiology	15	0.0
Traumatological and orthopedic surgery, neurosurgery, and neurology	5463	5.8
Hematology, oncology, nuclear medicine	988	6.4
Dermatology, general medicine1, infectious diseases (including COVID units), pulmonology, internal medicine, general medicine 2, rheumatology	4691	7.6
Emergency department	3568	0.8
Total H hospital	25,038	5.5
Hospital M		
Emergency department, Pediatrics and pediatric surgery, gynecology obstetrical, surgery	7628	2.7
ICU (surgical and medical including COVID-19)	1052	6.8
Total M hospital	8680	4.7
Total hospital H + M	33,718	5.4

**Table 2 healthcare-12-00046-t002:** Characteristics of patients and site of acquisition (modern versus historical hospital) according to nosocomial versus community-acquired COVID-19. Results are presented as n (%) and median (Q1 and Q3).

	Patients with Nosocomial COVID-19 (n = 150)
Male patients	85 (56.7%)
Female patients	65 (43.3%)
Age (years)	79 [69–85]
Death	49 (32.7%)
Delay of acquisition	12 d (7–19 days)
Single room (all hospital stay)	73 (48.6%)
Double room	77 (51.3%)
Secondary cases (double room)	26 (17.3%)
Modern hospital	1 (0.7%)
Hitorical hospital	149 (99.3%)

## Data Availability

The datasets generated and/or analyzed during the current study are available from the corresponding author on reasonable request.

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
