# Peer review of "Nosocomial Coronavirus Disease 2019 during 2020–2021: Role of Architecture and Ventilation"

_healthcare, 2023, doi:10.3390/healthcare12010046_

Round 1

Reviewer 1 Report (Previous Reviewer 1)

Comments and Suggestions for Authors

Accept in present form

Author Response

Thank you

Reviewer 2 Report (Previous Reviewer 2)

Comments and Suggestions for Authors

I stand by my comments with respect to "healthcare-2631413"
I am an architectural engineer by training, with some research works in the medical industry. According to my area of research, the methodology is weak due to the wide differences in design, ward areas, ACH between the old and new buildings. Hence, I maintain my position of rejecting this article. But, hospitals were built to serve a special purpose and these might not be very important with regards to the transmission of illness. This, I respectfully agree.

Comments on the Quality of English Language

NA

Author Response

As we said previously, we agree that H and M are very different, but this study aimed to compare very different healthcare structures to highlight the weight of architecture and ventilation. We have more precisely defined the characteristics of the buildings of H and M and carried out precise measurements of ACH in a sample of rooms of H and M. These measurements with a more precise description of the air flow of the rooms highlight the differences between the modern health facilities and old health facilities. and are important in explaining the role of collective facilities. This knowledge is important so that healthcare professionals can understand how nosocomial cases of COVID-19 arise and implement corrective measures.

We had one  old building and one new building with indeed wide differences in design, ward areas, ACH between but once again the aim was to compare them.   

Reviewer 3 Report (Previous Reviewer 3)

Comments and Suggestions for Authors

Hello,

My recommendations have been accepted and the article has been reformulated accordingly, so I have nothing more to add.

Congratulations on your work!

Author Response

Thank you 

Reviewer 4 Report (Previous Reviewer 4)

Comments and Suggestions for Authors

I am good with the reviewed version now. I believe the authors have put enough effort to make it more scientifically sound.

May be accepted for publication. 

Just on suggestion, after line 145, the table 1 should mention which column is for hospitalization numbers and which one is average length of stay. It's missing.

Author Response

Thank you for your remark. Table 1 was modified as suggested

This manuscript is a resubmission of an earlier submission. The following is a list of the peer review reports and author responses from that submission.

Round 1

Reviewer 1 Report

Comments and Suggestions for Authors

Martinot et al. described the role of architecture and ventilation in SARS-CoV-2 transmission. While virus transmission is influenced by many factors, under the condition of controlling variables, this research findings underscore the importance of adopting appropriate architectural and ventilation strategies in healthcare settings and public spaces to reduce virus transmission. These strategies can help lower the risk of infection during outbreaks and in everyday life. Some comments are as following:

1.Please provide a more detailed overview of the research related to the role of architecture and ventilation in virus transmission, especially the latest findings and strategies applied in healthcare settings.

2.In the discussion section, it is recommended to elaborate on specific strategies, such as the threshold for CO2 concentration monitoring, air filtration, airflow direction control, choice of building materials, etc. This will help readers better understand how to practically apply these principles to prevent airborne virus transmission.

3. In the discussion section, Emphasize the significance of architecture and ventilation during pandemic outbreaks and the necessity of implementing relevant measures in healthcare facilities and other public spaces.

4.Given the evolving nature of pandemics, consider mentioning areas that may require further research and improvements in the future to adapt to new challenges and changes.

Author Response

1.Please provide a more detailed overview of the research related to the role of architecture and ventilation in virus transmission, especially the latest findings and strategies applied in healthcare settings.

Thank you for your comment. We have added three new references underscoring the interest of new strategies and findings in healthcare settings both for NV and MV:

Megahed, N. A. and E. M. Ghoneim (2021). Indoor Air Quality: Rethinking rules of building design strategies in post-pandemic architecture." Environ Res 193: 110471.

Yang, Y. F., Y. J. Lin, S. H. You, T. H. Lu, C. Y. Chen, W. M. Wang and C. M. Liao (2023). "Control measure implications of COVID-19 infection in healthcare facilities reconsidered from human physiological and engineering aspects." Environ Sci Pollut Res Int 30(13): 36228-36243.

Zafari, Z., P. M. de Oliveira, S. Gkantonas, C. Ezeh and P. A. Muennig (2022). "The cost-effectiveness of standalone HEPA filtration units for the prevention of airborne SARS CoV-2 transmission." Cost Eff Resour Alloc 20(1): 22.

2.In the discussion section, it is recommended to elaborate on specific strategies, such as the threshold for CO2 concentration monitoring, air filtration, airflow direction control, choice of building materials, etc. This will help readers better understand how to practically apply these principles to prevent airborne virus transmission.

Thank you for pointing this out. Accordingly, we have revised the discussion section to integrate more specific strategies for HCWs for controlling aerosols, especially we introduced about monitoring CO2 thresholds :

These corrective methods could include integration of CO2 captors monitoring CO2 in medical departments allowing alerts for levels >700 ppm, for example (Piscitelli, Miani et al. 2022), alerting HCWs to increase ACH by opening the windows. Modelization, with the help of aerosol scientists, of the natural airflow within a department, and how doors and windows modified these airflows to control aerosol circulation is important to prevent far-field contamination and modify the location and care pathway for infected patients (Martinot 2023, Park, Yu et al. 2023). In case of low departments  with low ACH mobile filtration units, the use of UV-based technologies could be discussed.

  1. In the discussion section, Emphasize the significance of architecture and ventilation during pandemic outbreaks and the necessity of implementing relevant measures in healthcare facilities and other public spaces.

Thank you for your comment. We emphasized that these measures, although important in daily practice, are of utmost importance during outbreaks of pandemics. This point was added to the discussion section.

4.Given the evolving nature of pandemics, consider mentioning areas that may require further research and improvements in the future to adapt to new challenges and changes.

We agree with this point. For instance, the COVID-19 pandemic was modified by the high level of immunization among population and variants. Therefore, the most efficient collective protective equipment (MV with high ACH, HEPA mobile units, or UV-based technologies) should focus either on departments accepting patients who are highly susceptible, such as those who are highly immunosuppressed, those not indicated to receive immunization, or those who are critically ill as well as departments treating patients who are contagious, such as those in infectious disease wards (nephrology, graft etc.) or departments or collective zones with a high volume of inpatients (emergency wards or zones regrouping a high volume of inpatients in nursing homes).

Reviewer 2 Report

Comments and Suggestions for Authors

P1 and P2 should be highlighted in Figure 1.

The methodology is very weak. Results were insignificant.

Line 87-90: Wilcoxon rank sum test, Chi-square or Fisher’s exact test, Kaplan–Meier method were not presented in the results section.

H and M were not compared due to the sample size. How significant is the results then?

Table 1 is insignificant as the topic of this paper is NC and not just Covid-19.

The various setting measurement of ventilation (ACH, air speed etc) were not measured and the discussion were not carried out with respect to the common parameters of ventilation.

Single room (48.7%) and double room (51.3%) - no discussion were done with respect the room settings (distance between beds, direction of air flow etc).

Comments on the Quality of English Language

acceptable.

Author Response

P1 and P2 should be highlighted in Figure 1.

Thank you for this remark. The hospital M and historical hospital have been highlighted using two different colors.

The methodology is very weak. Results were insignificant.

This a retrospective study with a less robust methodology. However, in our view, these real-life results are very clear and independent of any statistics and highlight the interest of architecture or ventilation about which, most HCWs have very little knowledge in our view. We agree that this study has numerous biases, which is why we detailed these in the discussion section.

Line 87-90: Wilcoxon rank sum test, Chi-square or Fisher’s exact test, Kaplan–Meier method were not presented in the results section.

They  were used as stated in the method section, with Wilcoxon used to compare the continuous data (age) between NC and non-NC cases and Chi2 for categorical data (number of nosocomial cases between H and M). The Kaplan–Meier was used for the mortality curves.

H and M were not compared due to the sample size. How significant is the results then?

M and H were compared according to the total number of admissions and were thus significant (1 NC/8680 in M against 149/25038 Chi2 p < 0.00001). This point was indeed not clearly written in the result section; therefore, we have revised it accordingly.

Table 1 is insignificant as the topic of this paper is NC and not just Covid-19.

Table 1 shows that there were fewer admissions in M than in H and that the number of inpatients differed, indicating the presence of a bias in our study. However, we agree that Table 1 is less important; thus, we have simplified it.

The various setting measurement of ventilation (ACH, air speed etc) were not measured and the discussion were not carried out with respect to the common parameters of ventilation.

In fact, measuring ACH in our hospital was not possible; therefore, we had to rely on technical data given by our aerosol scientist and we agree that this is also a bias. However, the imprecise classification of NV and MV is also quite easily understandable by HCWs. We emphasized this point in the discussion section.

Single room (48.7%) and double room (51.3%) - no discussion were done with respect the room settings (distance between beds, direction of air flow etc).

Thank you for this remark. Air flow in buildings with NV is extremely dependent on opening of windows, door, and room settings. Unfortunately, in H there are a high number of buildings with different room sizes and the air flow patterns are difficult to describe. For instance, in one of the buildings of H, the single rooms have an area of 12 m² and double rooms have an area of 18,5 m²; however, in another building, single rooms have an area between 12 and 16 m², whereas the double rooms have an area between 19 and 26 m² . Furthermore, in M, single rooms have an area between 15 and 22 m², whereas the double rooms have an area of 24 m². We made the necessary changes in the revised manuscript to reflect this.

Reviewer 3 Report

Comments and Suggestions for Authors

Congratulations on your work.

 Studies similar to this one, carried out in different contexts, show that people's health is improved when they use personal and collective protective equipment.

In my opinion, the relevance of your work lies in the reinforcement that needs to be made to institutions and professionals in order to change practices.

Suggested change:

1.    The abstract should be revised.

2.    Readers should be informed in the methodology that the reference ACH was based on "technical" data, which is only realised later in the "Discussion".

3.    The definition of nosocomial COVID-19, which is different from the widely accepted one, should also be incorporated into the methodology.

4.    The text between lines 187-199 should be considered as "Conclusions"

I look forward to more studies that can overcome these limitations.

Author Response

Congratulations on your work.

 Studies similar to this one, carried out in different contexts, show that people's health is improved when they use personal and collective protective equipment.

In my opinion, the relevance of your work lies in the reinforcement that needs to be made to institutions and professionals in order to change practices.

Thank you for your remarks

Suggested change:

  1. The abstract should be revised.

We have revised the abstract to incorporate these remarks.

  1. Readers should be informed in the methodology that the reference ACH was based on "technical" data, which is only realised later in the "Discussion".

Thank you for these comments. The method for determining ACH has been described in the methods section.

  1. The definition of nosocomial COVID-19, which is different from the widely accepted one, should also be incorporated into the methodology.

The notion of the varying definition of NC was described in the methods section

  1. The text between lines 187-199 should be considered as "Conclusions"

We have included the concerned text as our “conclusions” section.

I look forward to more studies that can overcome these limitations.

We absolutely agree with this point.

Reviewer 4 Report

Comments and Suggestions for Authors

"Nosocomial coronavirus disease 2019 during 2020–2021: Role of 2 architecture and ventilation" is an attractive title in itself. Epidemiological studies to explore etiological factors facilitating COVID-19 spread is valuable for the clinical community.

The table-1 describing (line 109-110) could be simplifide for the reader to understand better. The heading is ambigous it seems. Needs a review.

If the study could be repeated prospectively, could help establising the hypothesis more better.

Anyway, it still is important towards establishing engireeing controls in IPAC measures in COVID-19 like pandemic situations.

Author Response

"Nosocomial coronavirus disease 2019 during 2020–2021: Role of 2 architecture and ventilation" is an attractive title in itself. Epidemiological studies to explore etiological factors facilitating COVID-19 spread is valuable for the clinical community.

Thank you for your valuable comment.

The table-1 describing (line 109-110) could be simplifide for the reader to understand better. The heading is ambigous it seems. Needs a review.

We completely agree with this remark and thus have simplified both table and title.

If the study could be repeated prospectively, could help establising the hypothesis more better.

Anyway, it still is important towards establishing engireeing controls in IPAC measures in COVID-19 like pandemic situations.

We agree that prospective and comparative studies are relatively much more difficult to elaborate.

Round 2

Reviewer 2 Report

Comments and Suggestions for Authors

I am an architectural engineer by training, with some research works in the medical industry. According to my area of research, the methodology is weak due to the wide differences in design, ward areas, ACH between the old and new buildings. Hence, I maintain my position of rejecting this article. But, hospitals were built to serve a special purpose and these might not be very important with regards to the transmission of illness. This, I respectfully agree.